# The Chain Mediating Role of Teachers’ Job Stress in the Influence of Distributed School Leadership on Job Satisfaction: Evidence from China, the United States, England, and Australia

**DOI:** 10.3390/bs14040279

**Published:** 2024-03-27

**Authors:** Jian Li, Eryong Xue, Yuxuan Liu

**Affiliations:** Institute of International and Comparative Education, Beijing Normal University, Beijing 100875, China; jianli209@bnu.edu.cn (J.L.); 202331010072@mail.bnu.edu.cn (Y.L.)

**Keywords:** distributed school leadership, teacher job satisfaction, teacher job stress, teacher leadership

## Abstract

Distributed leadership has been shown to improve teacher job satisfaction and reduce teacher job stress. However, few studies have thoroughly explored the indirect effects of distributed leadership on increasing the teachers’ burden in school administration and management, thereby increasing work stress, and decreasing job satisfaction. Data from the Teaching and Learning International Survey were analyzed to investigate the relationships among distributed school leadership, teachers’ job stress, and job satisfaction. A total of 3976 teachers from 198 junior high schools in Shanghai, 2560 teachers from 166 junior high schools in the United States, 2376 teachers from 157 junior high schools in England, and 3573 teachers from 238 junior high schools in Australia were selected and examined using structural equation modeling. The results revealed that distributed school leadership directly predicted teachers’ job satisfaction; teachers’ job stress had an independent mediating effect on distributed leadership and teachers’ job satisfaction, whereas teachers’ time spent participating in school leadership had no mediating effect. We discuss the benefits of distributed school leadership on teachers’ job satisfaction and the possible mechanisms for promoting it in practice.

## 1. Introduction

Distributed leadership has been reported to play a key role in teachers’ job satisfaction worldwide [1,2,3,4] The educational working environment can only be healthy and productive if teachers achieve an adequate level of job satisfaction [5]. However, the current situation regarding job satisfaction among primary and secondary school teachers in China is concerning. A survey conducted by Merrimack College in the United States found that only 12% of teachers in the United States said that they were “very satisfied” with their jobs [6]. Another study found that teachers in England in 2007 rated their job satisfaction significantly lower compared with teachers in 1962 [7,8]. Overall, job satisfaction among primary and secondary school teachers around the world is not high. Low job satisfaction of teachers is reported to be directly related to a high degree of work pressure [9]. Since Kyriaco and Sutclife first proposed the concept of teacher job stress in 1977 [10], teacher job stress has been examined in several studies. Additionally, the frequency of cases of extreme teacher burnout has brought this topic to the attention of education administrations [11,12]. Teachers’ job stress is derived from various factors, including economic hardship, working conditions, age, and personality traits, influencing teachers’ stress levels [1,2,3,4]. For example, according to the RAND Corporation’s “Teacher Happiness and Resignation Intention”, although the level of job stress of US teachers has recovered somewhat after the COVID-19 pandemic, the happiness of teachers is still not high compared with other workers. The main sources of job stress for US teachers are managing student behavior (46%), supporting students’ academic learning (34%), and administrative work (29%). About a quarter of teachers report that administrative work, low pay, and long hours are the biggest job-related stressors [13]. In November of the same year, Education Support—a British teacher welfare charity—released the Teacher Wellbeing Index 2023, which pointed out that the happiness of British teachers was at its lowest level since 2019, surveying more than 3004 educators. Among them, 78% of teachers reported burnout, up 3 percentage points from 2022, and 36% reported burnout, up 9 percentage points from 2022, with loneliness and social isolation as a source of stress [14]. According to the Mental Health Foundation of Australia, more than 50% of Australian teachers suffer from anxiety, and almost one in five suffer from depression [15]. It can be said that the increasing work pressure for teachers has become a reality in different countries [5]. The high work pressure caused by low income and heavy workload reduces the happiness of teachers and increases the turnover rate of teachers. Delegating power to teachers, adjusting workload freely, and relieving work pressure are the fundamental ways to increase teachers’ job satisfaction [9].

In this context, the distributed school leadership model, which involves the democratic participation of teachers, has quickly become a focus in school organization. Reforms of school organizational structure have urged school leaders to examine traditional hierarchical structures and distribute leadership functions more equally among all levels and staff members [16]. In primary and secondary schools, the traditional centralized leadership model has gradually been replaced by more decentralized and collaborative leadership [17], and this change reflects the pursuit of a more flexible, innovative, and participatory management model in educational institutions [18]. Under the framework of distributed leadership, teachers have more opportunities to participate in school decision-making and problem-solving processes. This sense of participation may make teachers feel more valued and recognized [19], thus improving work efficiency and satisfaction [20].

### 1.1. Distributed Leadership and Teachers’ Job Satisfaction

Distributed school leadership is a leadership model in which different subjects inside and outside the school participate in the school management process. The concept of distributed leadership can be traced back to Gibb’s *Handbook of Social Psychology* from 1954 [21], and more systematic and in-depth research was conducted by the British Institute of Educational Leadership, Management and Administration in the early 2000s. Distributed leadership is the most frequently repeated topic in the conference communication papers catalog [22]. Gronn divides distributed leadership into quantitative behavior and collaborative behavior, which emphasizes two important characteristics of distributed leadership [23]. Distributed leadership can also be defined as a shared practice of decision making involving multiple levels, including leaders, followers, and scenarios involving leaders and followers interacting with each other over time [24]. Harris pointed out that the core of the concept of distributed leadership is that leadership is not owned by any one person but is a function, which all members of the organization may play [20]. Accordingly, leadership is not fixed but fluid, generative, and changing, and teachers, experts, students, and parents can participate in the practice of school leadership. Based on the Teaching and Learning International Survey (TALIS), the current study adopted a narrow definition and regarded distributed leadership as a behavioral practice in which principals authorize teachers to participate in school organizational decision making [25]. We focused on the group leadership behavior of teachers under the distributed leadership model of schools. The TALIS defines the actual performance of distributed leadership as an opportunity to participate in school decision making, the need to take responsibility for school affairs, and the ability to feel the cooperative cultural atmosphere. This is the degree of distributed leadership in teachers’ subjective perception, and it is not an objective principal leadership style or distributed leadership atmosphere on campus.

Job satisfaction generally refers to an individual’s subjective evaluation of their work and positive emotional state. It relates to the sense of accomplishment and achievement generated by everyday activities of the profession [26,27]. There are many kinds of external factors, which also significantly impact job satisfaction and occupational stress. For example, Din-ham and Scott classify the sources of teacher job satisfaction into “core business of teaching” factors (student achievement, professional self-growth); school-level factors (school leadership, climate, decision making, school infrastructure, school reputation); and system-level/social factors (workload and impact of change, the status and image of teachers, promotion results) [28]. The principal’s behavior creates a different working environment in the school, which in turn affects teacher satisfaction. In the field of education, teachers’ job satisfaction refers to the satisfaction experienced by teachers in the process of education and teaching [29]. Teachers’ job satisfaction has a multi-dimensional structure. Teachers who are satisfied with their work are likely to move to other schools or leave the teaching team because of the inadequacy of their school environment [30]. Distributed leadership is an important factor affecting teachers’ job satisfaction [31]. When leadership activities are widely distributed among teachers, teachers can play the role of leaders, assume corresponding leadership functions, and participate in the establishment of school development goals, which can effectively improve teachers’ job satisfaction and work enthusiasm [32]. The value-added effects and promoting mechanisms of distributed leadership have been confirmed in empirical studies [33].

### 1.2. Teachers’ Work Pressure and School Leadership Participation Time

Different types of working hours have different impacts on teachers’ working status. Kong et al. divided the working time of primary and secondary school teachers into teaching working time and non-teaching working time, based on existing studies [34]. Teaching working time refers to working time, which is closely related to teaching. Non-teaching working time refers to working time, which is not directly related to teaching but is necessary to maintain the teaching order and improve the teaching level. Relative to power and responsibility, distributed school leadership, while empowering teachers, also means transferring leadership tasks to teachers, which will naturally increase the proportion of teachers’ working time spent on school management and daily administration, which is part of non-teaching time. Based on a case study of four primary school teachers, Ballet et al. found that increasing teachers’ non-teaching tasks and paying less attention to professional behaviors can lead to professional degradation and decrease teachers’ sense of professional identity [35]. However, based on TALIS 2018 data, Sun reported that neither school management time nor general administrative work time had a significant impact on teachers’ job satisfaction [36]; therefore, the relationship between the two needs to be further explored. Teacher stress is regarded as the experience of unpleasant and negative emotions arising from teachers’ daily work [26,37]. Teachers’ pressure sources can be divided into exogenous and endogenous sources. Exogenous sources include student behavior and achievement, working environment, salary and school support, and stakeholders’ requirements, etc., while endogenous sources include teachers’ personality characteristics, interpersonal conflicts, and self-expectations [38]. Factors affecting teachers’ stress can be generally divided into external environmental factors and internal personal factors. Academic circles pay more attention to environmental factors, such as time pressure, student behavior, working conditions, leadership support level, school atmosphere, etc. [10,39,40,41], but less attention has been paid to individual factors, including subjective wellbeing [42], cognitive assessment [43], and personality factors [44]. Autonomy-supportive leadership factors [45] and distributed leadership will reduce teachers’ work pressure [46]. However, there is a negative correlation between teachers’ job stress and teachers’ job satisfaction [47]. Reducing work stress will improve teachers’ job satisfaction. For knowledge employees, work pressure is an important variable affecting job satisfaction, and an increase in work pressure leads to a decline in job satisfaction [48].

The specific work tasks completed by teachers in performing their professional roles and the time required indicate the workload of teachers, including the total amount of work time and the time allocation of specific work tasks [49]. Lack of control over work and workload, blurring of boundaries between work and home, pressure from online teaching, irregular working hours, and financial problems can increase work stress for teachers [50]. Zhou et al. found that among exogenous stressors, workload had the greatest impact on teachers’ stress [38]. Therefore, the dissemination of the distributed leadership model in the actual teaching management process increases the non-teaching tasks of teachers, resulting in an increase in teachers’ work pressure and job satisfaction [49].

In a study of Irish primary school teachers, researchers reported that school members’ understanding of distributed leadership involved sharing responsibility and workload rather than leadership itself. From this perspective, leadership is shared only among those who already hold leadership positions, not “outsiders”. The way leadership is shared is primarily a phenomenon of delegating authority (passive commitment) rather than encouraging participation (active acquisition) [51]. Distributed leadership involves additional workload for teachers. Regardless of whether the increase in workload is active or passive, this increment cannot be ignored. Although educational administration departments and schools support the policy of carrying out distributed leadership, it has been reported that many staff believe that distributed leadership increases the work burden [51] and that the implementation of distributed leadership does not consider real-world situations and cannot achieve the desired effects.

There is no consensus on whether distributed leadership improves teachers’ job satisfaction or not. Yokota found that different social and cultural factors, as well as national political forces, are reported to hinder the development of distributed leadership [52], and there is a gap between the practice of distributed leadership and policy expectations [53]. This also shows that the relationship between distributed leadership and teachers’ job satisfaction can be further investigated from the perspective of country comparison. Although more studies have examined teacher stress and satisfaction since the outbreak of the coronavirus disease 2019, few studies have examined the impact of distributed school leadership on teacher stress and satisfaction from an international perspective. 

The current study aimed to explore the relationship between distributed leadership and teachers’ job satisfaction through the intermediary of teachers’ job stress and the proportion of teachers’ total work time spent participating in school leadership. The current study used TALIS 2018 data from regions in four continents to examine the mechanisms by which distributed school leadership affects teacher job satisfaction. The hypothesized relationships between these factors are presented in Figure 1. The research questions were as follows:

Q1: What role does teachers’ job stress play in the relationship between distributed leadership and teachers’ job satisfaction?

Q2: What role does teachers’ time spent participating in leadership play in the relationship between distributed leadership and teachers’ job satisfaction?

Q3: Does the time spent by teachers participating in leadership and teachers’ job stress have a chain mediating effect between distributed leadership and teachers’ job satisfaction? 

## 2. Methods

### 2.1. Data Source and Sample Composition 

In this study, SPSS 27.0 was used for reliability testing, descriptive statistics, and correlation analysis, and AMOS21.0 was used for structural equation model testing and bootstrap mediation effect testing. At the end of 2017, TALIS surveyed approximately 240,000 teachers and 13,000 head teachers in nearly 50 regions, enabling the data to be used for comparative studies across different countries [25]. Yokota reported that the traditional role of the principal arising from social, cultural, and political forces hinders the development of the distributed leadership model [52]. In a study conducted in Malaysia, Bush concluded that the implementation of distributed leadership is rooted in cultural norms rather than policy regulations and that it is not always possible to achieve distributed leadership through theoretical deduction [54]. To compare the mechanisms of influence of the differences in distributed school leadership on teachers’ job satisfaction in different regions, the current study examined four regions in Asia, Europe, America, and Oceania, which exhibit cultural differences and regional representation and have excellent results in the OECD PISA test (Shanghai, England, the United States, and Australia) for comparison.

We examined data from the TALIS conducted by the OECD in 2018. The study aimed to investigate the basic conditions, teaching environment, and working status of principals and teachers in sample schools by conducting questionnaires with principals and teachers from different regions to examine the problems affecting schools and the future career development of teachers. A two-stage stratified sampling method was used to investigate the basic conditions, teaching environment, and working status of principals and teachers in sample schools [25]. The PPS sampling method was adopted in all TALIS samples to ensure that the samples have good representativeness.

In the current study, the sample data of junior high school teachers in Shanghai, England, the United States, and Australia were selected. The original data from Shanghai included 3976 teachers from 198 junior high schools; data from the United States included 2560 teachers from 166 junior high schools; data from England included 2376 teachers from 157 junior high schools; and data from Australia included 3573 teachers from 238 junior high schools. Samples with missing values and outliers in the core variables were deleted from the four groups of samples, and the basic information for the final four samples is shown in Table 1. 

### 2.2. Variables and Measurements

The TALIS measures distributed leadership according to the concept proposed by Hallinger [55]. According to the TALIS technical report, the distributed leadership variable is reflected in the “stakeholder participation” dimension (code: T3PLEADP), and three related questions were included in the teacher questionnaire (recorded as DL1-3 in this study), such as “Schools provide teachers with the opportunity to actively participate in school decision making”. The current study mainly focused on three questions in the teacher questionnaire. Responses in the TALIS were measured on a 4-point Likert scale, ranging from 1 (strongly disagree) to 4 (strongly agree).

Based on the selected data, a reliability test was conducted on distributed leadership in the four countries. All Cronbach’s alpha coefficients were greater than 0.8.

Teachers are involved in school management and daily administrative tasks. According to the TALIS questionnaire, teacher work time includes the total work time of the most recent week, teaching time, time spent on lesson preparation, teaching, research (communication and cooperation with colleagues), homework correction, tutoring students, school management, daily administrative work (including oral communication, written work, and other paperwork), professional development activities, communication with parents, extracurricular activities, and other work time. Twelve questions in the current study focused on teachers’ answers to two blank-filling questions—“participation in school management” (code: TT3G18E) and “daily administrative working time” (code: TT3G18F)—and the proportions of the two were calculated based on total working time.

The TALIS divides teachers’ work stress into four dimensions, with a total of fifteen questions: workplace stress (code: T3WELS), including four questions (one reverse question, recorded as WPS1-4 in this study), such as “I feel stressed at work”; workload stress (code: T3PWLOAD), with a total of five questions (WLS1-5 in this study), such as “too much administrative work to do”; stakeholder behavioral stress (code: T3STBEH), with a total of six topics (recorded in this study as SBS1-3 and SS1-3), such as “to be responsible for student achievement”, “to keep up with the changing requirements of the district, city, or national education administration”. Based on selected data, reliability testing was conducted on the three dimensions of teachers’ job stress in the four countries. Cronbach’s alpha coefficient is shown in the table below. The reliability of stakeholder behavioral pressure in British classrooms was slightly lower, but the overall reliability of the four scales was above 0.8.

The TALIS divides teacher job satisfaction (code: T3JOBSA) into two dimensions: work environment satisfaction (code: T3JSENV) and teacher career satisfaction (code: T3JSPRO), with a total of 10 questions. Work environment satisfaction refers to teachers’ satisfaction with the working environment of their school, including three questions (one reverse question, recorded as JSE1-3 in this study), such as “I like working in this school”; career satisfaction refers to teachers’ satisfaction with the teaching profession, including six questions (two reverse questions, recorded as JEP1-6 in this study), such as “the advantages of being a teacher obviously outweigh the disadvantages”; the survey also includes a question on overall satisfaction. Based on the selected data, the reliability test was conducted on the two dimensions of teachers’ job satisfaction in the four countries. Cronbach’s alpha coefficients are shown in the table below, all of which are above 0.7, and Cronbach’s alpha coefficient of the whole scale is above 0.8.

### 2.3. Cross-Cultural Invariance

In cross-cultural research, the scales of different variables are translated into many versions to test subjects from different countries; therefore, it is necessary to determine whether the scales of variables are consistent in different measurements through measurement invariance analysis [56]. The data source for this study is a survey conducted by TALIS in 2018. The results of the cross-country measurement invariance of the questionnaire are published in the TALIS2018 Technical Report of the OECD, and the invariance hierarchy is divided into three levels [57]: the configural level (the factor loadings and intercepts are allowed to vary across participating countries/economies; further analysis of cross-country comparisons can only be carried out at the conceptual level), the metric level (the score of the scale is created, respectively, with equal factor loadings but with intercepts allowed to vary across participating countries/economies, which can be used for correlation- and linear-regression-based analyses), and the scalar level (which can compare the average size of participating countries/economies). Two dimensions of job stress and two dimensions of job satisfaction were included. At the same time, Mplus software 8.7 as used to analyze the measurement invariance of distributed leadership variables and stakeholder dimensions of job stress, and the same rating method was used in the technical report. The invariance level analysis results of each dimension of the final three variables are shown in the table below. Although each variable does not reach the scalar level, correlation and linear regression analyses can be carried out (Table 2).

## 3. Results

### 3.1. Descriptive Statistics and Discriminative Validity Test of Variables

The descriptive statistics and relevant analysis of distributed leadership, teachers’ participation time in school leadership, teachers’ job pressure, and teachers’ job satisfaction in the four countries are shown in the following table. In the Shanghai sample, distributed leadership was significantly positively correlated with teachers’ proportion of time spent participating in school leadership (r = 0.082, *p* = 0), working environment satisfaction (r = 0.531, *p* = 0), career satisfaction (r = 0.418, *p* = 0), and overall job satisfaction (r = 0.345, *p* = 0). Distributed leadership was significantly negatively correlated with workplace stress (r = −0.265, *p* = 0), workload pressure (r = −0.227, *p* = 0), and stakeholder behavioral pressure (r = −0.164, *p* = 0). All three kinds of stress were significantly negatively correlated with work environment satisfaction, career satisfaction, and overall job satisfaction of teachers, but only stakeholder behavioral stress was significantly negatively correlated with teachers’ proportion of time spent participating in school leadership (r = −0.043, *p* = 0). Teachers’ participation time in school leadership was significantly positively correlated with work environment satisfaction (r = 0.068, *p* = 0) and career satisfaction (r = 0.049, *p* = 0.012).

In the United States sample, distributed leadership was significantly positively correlated with work environment satisfaction (r = 0.523, *p* = 0), career satisfaction (r = 0.262, *p* = 0), and overall teacher job satisfaction (r = 0.369, *p* = 0), and it was significantly negatively correlated with workplace stress (r = −0.263, *p* = 0), workload pressure (r = −0.152, *p* = 0), and stakeholder behavioral pressure (r = −0.192, *p* = 0). The three kinds of stress were negatively correlated with teachers’ work environment satisfaction, career satisfaction, and overall job satisfaction. Teachers’ time spent participating in school leadership was positively correlated with career satisfaction (r = 0.052, *p* = 0.028).

In the sample from England, distributed leadership was significantly positively correlated with the proportion of time teachers spent participating in school leadership (r = 0.087, *p* = 0), working environment satisfaction (r = 0.556, *p* = 0), career satisfaction (r = 0.350, *p* = 0), and overall job satisfaction of teachers (r = 0.406, *p* = 0). Distributed leadership was significantly negatively correlated with workplace stress (r = −0.342, *p* = 0), workload pressure (r = −0.255, *p* = 0), and stakeholder behavioral pressure (r = −0.216, *p* = 0). The three kinds of stress were significantly negatively correlated with work environment satisfaction, career satisfaction, and overall job satisfaction of teachers, but only workload stress was significantly negatively correlated with teachers’ time spent participating in school leadership (r = −0.116, *p* = 0).

In the sample from Australia, distributed leadership was significantly positively correlated with teachers’ work environment satisfaction (r = 0.524, *p* = 0), career satisfaction (r = 0.326, *p* = 0), and overall job satisfaction (r = 0.390, *p* = 0), and it was significantly negatively correlated with workplace stress (r = −0.309, *p* = 0), workload pressure (r = −0.225, *p* = 0), and stakeholder behavioral pressure (r = −0.223, *p* = 0). The three kinds of stress were significantly negatively correlated with work environment satisfaction, career satisfaction, and overall job satisfaction of teachers, and teachers’ time spent participating in school leadership was significantly positively correlated with workplace stress (r = 0.061, *p* = 0) and significantly negatively correlated with workload pressure (r = −0.046, *p* = 0) and stakeholder behavioral pressure (r = −0.041, *p* = 0). The significant correlations between different country variables were in agreement with theoretical expectations and were further analyzed (Table 3).

Additionally, confirmatory factor analysis was used to evaluate the discriminative validity of the variables. The four variables in this study were combined to construct one four-factor model, three three-factor models, one two-factor model, and one single-factor model. The results are shown in the table below. Compared with other alternative measurement models (Models 1–4), the degree of fit of the four countries was not ideal. These results indicated that the model of distributed school leadership, teachers’ time spent participating in school leadership, teachers’ job pressure, and teachers’ job satisfaction was not valid (Table 4).

The purpose of Models 1–5 is to show that the four variables belong to different factors, but for TALIS data, the “duration of teachers’ participation in school management and administrative affairs” is the explicit variable, and only one blank-filling question seriously affects the result of discrimination validity analysis. Considering that the time proportion of teachers’ participation in school leadership itself was an explicit variable, and the validity of the topic itself was limited, the benchmark model was adjusted to exclude the variable of time proportion of teachers’ participation in school leadership, and a three-factor model, a two-factor model, and a single-factor model were constructed. Compared with other alternative measurement models (Model 1 and Model 2), the three-factor model (benchmark model) showed a better fit, indicating that distributed school leadership, teachers’ job stress, and teachers’ job satisfaction are four different constructs and that the model had good discriminative validity, showing that distributed leadership, job satisfaction, and job stress are different variables and can be clearly distinguished (Table 5).

In this study, AMOS 21.0 was used to test the fit degree of the hypothesis model through the structural equation model, and the mediation effect was further analyzed with the bootstrap test. First, we constructed Model A in which distributed school leadership affected teachers’ job satisfaction through teachers’ job stress, and distributed school leadership predicted teachers’ job satisfaction. Second, we constructed Model B in which distributed school leadership affected teachers’ job satisfaction through the proportion of teachers’ time spent participating in school leadership, and distributed school leadership predicted teachers’ job satisfaction. Finally, a chain mediating role model (Model C) was constructed between teachers’ job stress and the proportion of teachers’ time spent participating in school leadership and teachers’ job satisfaction (Table 5).

### 3.2. The Relationships between Distributed School Leadership and Teachers’ Job Satisfaction

Distributed school leadership significantly positively predicted teachers’ job satisfaction (β = 0.388, *p* < 0.01), significantly negatively predicted teachers’ job stress (β = −0.265, *p* < 0.01), and significantly negatively predicted teachers’ job satisfaction (β = −0.393, *p* < 0.01). Bootstrapping was further used to test the partial mediating effect of teachers’ job stress. Bootstrap resampling was set to 5000 times, and 95% confidence intervals (CIs) were calculated. The analysis results revealed that the direct effect of distributed school leadership on teachers’ job satisfaction was significant (95% CI: 0.349, 0.428), and the direct effect value was 0.388. The mediating effect of teachers’ job stress was significant (95% CI: 0.083, 0.128), and the mediating effect value was 0.104, accounting for 21% of the total effect. Therefore, teachers’ job stress plays a partial mediating role between distributed school leadership and teachers’ job satisfaction.

Distributed school leadership significantly positively predicted the proportion of time teachers spent participating in school leadership (β = 0.015, *p* < 0.01) and teachers’ job satisfaction (β = 0.454, *p* < 0.01), but the proportion of time teachers spent participating in school leadership did not significantly predict teachers’ job satisfaction (β = 0.109, *p* = 0.142). Bootstrapping was further used to test the partial mediating effect of teachers’ time spent participating in school leadership. Therefore, the results indicated that the proportion of time teachers spent participating in school leadership did not play a partial mediating role between distributed school leadership and teachers’ job satisfaction, and Hypothesis 3 was not supported. Finally, a chain intermediary model of distributed school leadership, teachers’ work pressure, teachers’ time spent participating in school leadership, and teachers’ job satisfaction was constructed. 

The results showed that distributed leadership still had a significant positive predictive effect on teachers’ job satisfaction when the two mediating variables of teachers’ job stress and teachers’ time spent participating in school leadership were included (β = 0.385, *p* < 0.01). Thus, Hypothesis 1 was supported. The results of the bootstrap mediation effect test showed that the total indirect effect of the model was (−0.27) × (−0.39) + 0.02 × 0.12 + 0.02 × 0.02 × (−0.39) = 0.11, accounting for 21.86% of the total effect. The independent mediating effect value of teachers’ work stress was 0.104 (95% CI: 0.082, 0.128), and the independent mediating effect value of teachers’ time spent participating in school leadership was 0.02 (95% CI: 0.000, 0.005). Finally, a chain intermediary model of distributed school leadership, teachers’ work pressure, teachers’ time spent participating in school leadership, and teachers’ job satisfaction in the United States, England, and Australia was constructed (Table 6 and Figure 2).

## 4. Discussion

In the current study, structural equation model testing and bootstrap analysis were conducted using the TALIS 2018 junior high school teacher data from Shanghai, the United States, England, and Australia, and the relationship between teachers perceived distributed school leadership and teachers’ job satisfaction was explored, as well as the mediating role of teachers’ job stress and teachers’ time spent participating in school leadership. 

Distributed school leadership had a significant positive predictive effect on teachers’ job satisfaction in different countries. 

Although active participation in school leadership increases the proportion of non-teaching time, the impact on work stress is not significant, which can be explained by teachers’ work initiative and enthusiasm; it can also be explained by the low proportion of school leadership time in the four countries (the average value is around 10%). Ordinary teachers do not really participate in school leadership in ordinary times, and their energy consumption is insignificant compared with lesson preparation, teaching and research, homework correction, and communication with parents, which are not sufficient to make ordinary teachers really feel the pressure or even to have a direct impact on job satisfaction. Therefore, in the model, there is no intermediary role [12,31,58,59].

It is found that the four countries selected have commonalities, but the model ultimately reveals differences among the four countries. For example, the results of the study show that distributed leadership in Shanghai and England has a significant positive impact on the amount of time teachers spend participating in school leadership, while distributed leadership in the United States and Australia does not have such effect. The education systems in England and China are more centralized than those in the other two countries, with a tradition of top-down decision making and communication in education management. Distributed leadership is indeed an obvious and mandatory requirement for teachers to become school leaders, and teachers’ identity increases, their responsibilities increase, and their working hours increase significantly. In Shanghai and the United States, the proportion of teachers’ time spent participating in school leadership has a positive impact on teachers’ job satisfaction, while in England and Australia, it has a negative impact on teachers’ job satisfaction. This reflects the two different teacher work cultures, which believe that “more effort will produce more return”. In Shanghai and Australia, the proportion of teachers’ time spent participating in school leadership has a positive impact on teachers’ stress, while in Britain and the United States, it has a negative impact on teachers’ stress, which reflects that teachers in Britain and the United States may have a higher ability to withstand pressure or have more love for the teaching profession [1,2,3]. In addition, teachers’ job stress played an independent mediating role, while teachers’ time spent participating in school leadership did not. Teachers’ job stress did not obscure the relationship between distributed school leadership and teachers’ job satisfaction but further promoted the relationship. Distributed leadership can reduce teachers’ workload pressure and thus improve teachers’ organizational commitment [4,11,12].

## 5. Limitations and Prospects for Future Research

The current study involved several limitations, which should be considered. First, because the study only included teacher data from the TALIS 2018 in Shanghai, the United States, England, and Australia, the representativeness of the data for populations in Asia, America, Europe, and Oceania may have been biased. Additionally, the research results may not be representative of the effects of distributed leadership in promoting teachers’ job satisfaction in other regions, limiting the extensibility of our research conclusions. Follow-up studies should consider cluster sampling in different continents based on economic status, population, and other factors; collect more extensive sample data; and compare and discuss the practical effects of distributed leadership in different schools and regional differences in depth. Second, the proportion of teachers’ time spent participating in school leadership was the sum of the time spent on “participation in school management” and “daily administrative work” reported by teachers and the proportion of “total working time” as the explicit variable, and the validity of the topic was not high, reducing the overall explanatory power of the structural equation model constructed in the study. Future research should classify different categories of teachers’ working hours and set them as control variables for regression analysis. Third, teachers’ subjective perceptions of distributed school leadership were examined in the questionnaire, and the relevant variables of “stakeholder participation in school management” in the questionnaire from the principals’ perspective were not considered. In follow-up research, a multi-layer linear model analysis method should be adopted to further incorporate principals’ perceptions regarding distributed school leadership into the model. Finally, distributed school leadership is an interactive behavior between principals and teachers, and teachers’ attitudes and behaviors may change according to differences in leaders’ behaviors and actual working situations, which cannot be captured from cross-sectional data. Future studies should consider using longitudinal tracking and experiments to further explore the impact of distributed school leadership on teachers’ job satisfaction. Fourth, in addition to teacher leadership behavior, other external factors—including exogenous factors, such as the economy, education policy, and workload, as well as endogenous factors, such as gender, teaching experience, and organizational commitment—can also significantly affect teachers’ job satisfaction and work stress. These factors are not considered in the summary of this study. The mechanism of these factors can be further discussed in the future. Fifth, in terms of analysis methods, this study mainly adopts AMOS path analysis and does not deeply explore the mediating role of variables (and possible moderating role).

## Figures and Tables

**Figure 1 behavsci-14-00279-f001:**
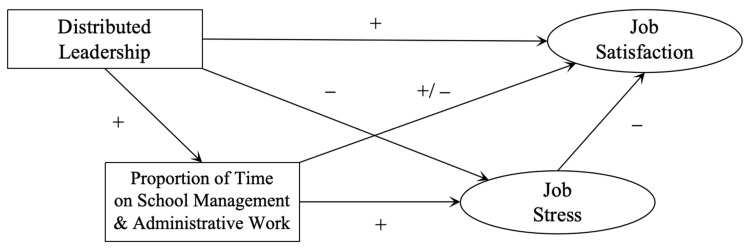
Hypothesized relationship path diagram. Note: + indicates a positive impact; − indicates a negative impact; and +/− indicates an uncertain impact direction.

**Figure 2 behavsci-14-00279-f002:**
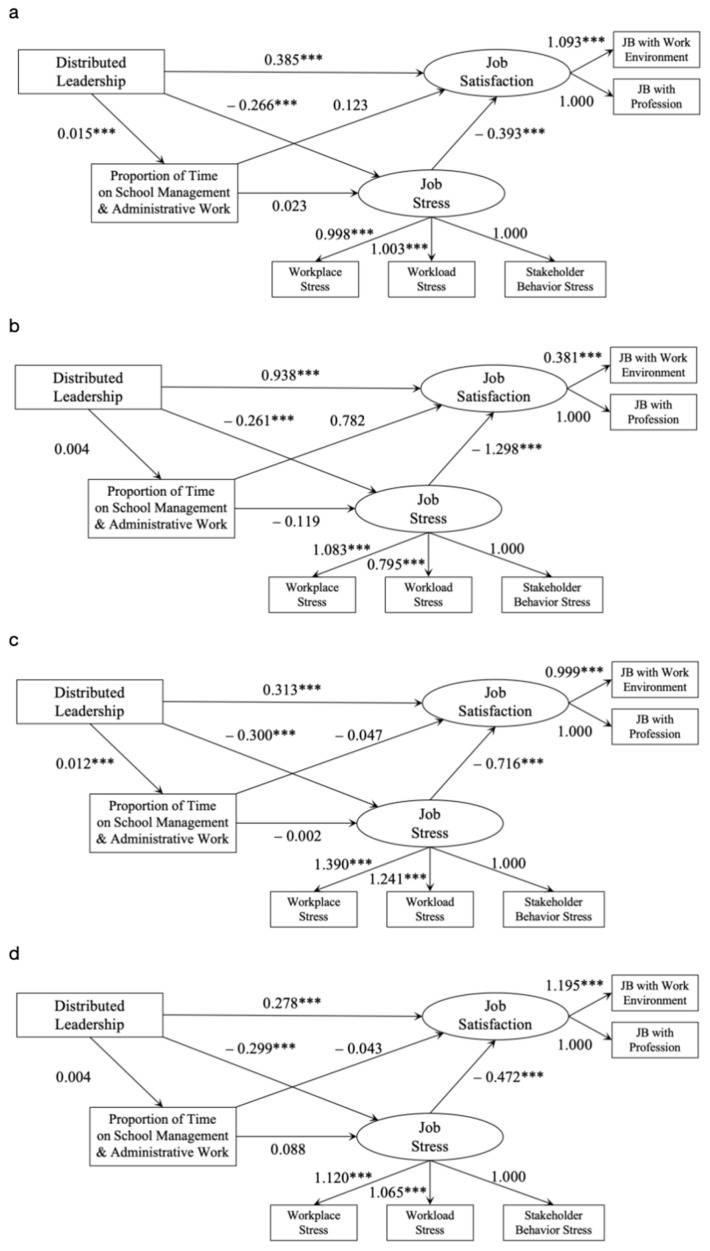
Chain mediation model of the effects of distributed school leadership on teachers’ job satisfaction. Notes: (**a**) represents Shanghai (China); (**b**) represents the United States; (**c**) represents England (United Kingdom); (**d**) represents Australia. Note: *** represents the standardized estimates significant at 0.001 level.

**Table 1 behavsci-14-00279-t001:** Basic information for teacher samples in four countries.

Variable	Number of Teachers in Shanghai	Number of Teachers in the US	Number of Teachers in the UK	Number of Teachers in Australia
Total (collected)	3976	2560	2376	3573
Total	2664	1758	1646	2333
Teacher gender				
Female	1958	1197	1049	1492
Male	706	561	597	841
Teacher education				
Bachelor’s degree or below	2293	646	1232	1895
Graduate student	371	1112	414	438
School type				
Civilian run	436	117	1059	825
Government-run	2228	1641	587	1508
School location (after basic processing)				
Town and village	325	546	413	290
City	2339	1212	1233	2043

**Table 2 behavsci-14-00279-t002:** Invariance test results for each scale.

Scale Label	Invariance Level	RMSEA	CFI	TLI	SRMR	∆CFI	∆TLI
Job satisfaction with work environment, teacher	Metric	0.056	0.969	0.962	0.092	0.017	−0.005
Job satisfaction with profession, teacher	Metric	0.071	0.961	0.94	0.088	0.037	0.049
Workload stress	Metric	0.068	0.949	0.936	0.069	0.027	0.003
Workplace stress	Metric	0.088	0.948	0.936	0.09	0.035	0.012
Stakeholder behavioral stress	Metric	0.079	0.958	0.94	0.089	0.035	0.015
Distributed leadership	Metric	0.093	0.995	0.998	0.044	0.025	0.023

**Table 3 behavsci-14-00279-t003:** Mean values, standard deviations, and correlation coefficients of variables in four countries.

Shanghai (China)												
	N	M	SD		T	DL	WPS	WLS	SS	JSE	JSP	JSO
T	2664	0.0824	0.1201		--							
DL	2664	3.3419	0.64493	r	0.082 **	--						
				*p*	0							
WPS	2664	2.747	0.76007	r	−0.008	−0.265 **	--					
				*p*	0.685	0						
WLS	2664	2.4572	0.75787	r	0.002	−0.227 **	0.560 **	--				
				*p*	0.938	0	0					
SS	2664	2.9751	0.80455	r	−0.043 *	−0.164 **	0.527 **	0.591 **	--			
				*p*	0.027	0	0	0				
JSE	2664	3.5038	0.67559	r	0.068 **	0.531 **	−0.438 **	−0.322 **	−0.261 **	--		
				*p*	0	0	0	0	0			
JSP	2664	4.2021	0.70233	r	0.049 *	0.418 **	−0.445 **	−0.315 **	−0.307 **	0.653 **	--	
				*p*	0.012	0	0	0	0	0		
JSO	2664	3.11	0.574	r	0.028	0.345 **	−0.288 **	−0.198 **	−0.177 **	0.548 **	0.620 **	--
				*p*	0.147	0	0	0	0	0	0	
United States												
	N	M	SD		T	DL	WPS	WLS	SS	JSE	JSP	JSO
T	1758	0.0566	0.06759		--							
DL	1758	3.2061	0.70566	r	0.037	--						
				*p*	0.124							
WPS	1758	2.7823	0.83141	r	−0.005	−0.263 **	--					
				*p*	0.849	0						
WLS	1758	2.4514	0.78331	r	−0.018	−0.152 **	0.522 **	--				
				*p*	0.44	0	0					
SS	1758	3.1289	0.89172	r	−0.04	−0.192 **	0.567 **	0.494 **	--			
				*p*	0.091	0	0	0				
JSE	1758	3.7153	0.73437	r	0.043	0.523 **	−0.407 **	−0.247 **	−0.321 **	--		
				*p*	0.072	0	0	0	0			
JSP	1758	11.4253	2.46434	r	0.052 *	0.262 **	−0.462 **	−0.285 **	−0.366 **	0.439 **	--	
				*p*	0.028	0	0	0	0	0		
JSO	1758	3.17	0.668	r	0.033	0.369 **	−0.492 **	−0.304 **	−0.350 **	0.583 **	0.671 **	--
				*p*	0.164	0	0	0	0	0	0	
England (United Kingdom)												
	N	M	SD		T	DL	WPS	WLS	SS	JSE	JSP	JSO
T	1646	0.1027	0.09356		--							
DL	1646	3.1548	0.68925	r	0.087 **	--						
				*p*	0							
WPS	1646	3.2201	0.90489	r	0.032	−0.342 **	--					
				*p*	0.196	0						
WLS	1646	3.69	0.95797	r	−0.116 **	−0.255 **	0.548 **	--				
				*p*	0	0	0					
SS	1646	3.2875	0.81508	r	−0.042	−0.216 **	0.498 **	0.492 **	--			
				*p*	0.088	0	0	0				
JSE	1646	3.4942	0.7509	r	0.047	0.556 **	−0.471 **	−0.331 **	−0.345 **	--		
				*p*	0.057	0	0	0	0			
JSP	1646	3.6967	0.81449	r	0.02	0.350 **	−0.541 **	−0.418 **	−0.423 **	0.532 **	--	
				*p*	0.407	0	0	0	0	0		
JSO	1646	2.87	0.701	r	0.021	0.406 **	−0.538 **	−0.365 **	−0.384 **	0.635 **	0.672 **	--
				*p*	0.405	0	0	0	0	0	0	
Australia												
	N	M	SD		T	DL	WPS	WLS	SS	JSE	JSP	JSO
T	2333	0.1141	0.11181		--							
DL	2333	3.1538	0.72168	r	0.026	--						
				*p*	0.217							
WPS	2333	2.961	0.84009	r	0.061 **	−0.309 **	--					
				*p*	0.003	0						
WLS	2333	3.1495	0.94048	r	−0.046 *	−0.225 **	0.522 **	--				
				*p*	0.026	0	0					
SS	2333	3.1071	0.89875	r	−0.041 *	−0.223 **	0.485 **	0.493 **	--			
				*p*	0.047	0	0	0				
JSE	2333	3.5706	0.76379	r	0.007	0.524 **	−0.436 **	−0.253 **	−0.320 **	--		
				*p*	0.753	0	0	0	0			
JSP	2333	4.1515	0.7417	r	−0.017	0.326 **	−0.460 **	−0.353 **	−0.370 **	0.524 **	--	
				*p*	0.404	0	0	0	0	0		
JSO	2333	3.12	0.63	r	−0.002	0.390 **	−0.462 **	−0.290 **	−0.338 **	0.635 **	0.658 **	--
				*p*	0.934	0	0	0	0	0	0	

Notes: **. At a level of 0.01 (two-tailed), the correlation was significant. *. At a level of 0.05 (two-tailed), the correlation was significant. T represents the proportion of teachers’ time spent participating in school leadership; DL represents the city leaders of the school distribution; WPS represents the pressure of teachers’ workplace; WLS represents the pressure of teachers’ workload; SS represents the pressure of stakeholders; JSE represents teachers’ working environment satisfaction; JSP represents teachers’ career satisfaction; and JSO represents teachers’ overall satisfaction.

**Table 4 behavsci-14-00279-t004:** Comparison of four measurement models.

Shanghai (China)								
Model	Δχ^2^	Δdf	χ^2^/df	*p*	RMSEA	NFI	CFI	GFI
Reference model	213.033	10	21.303	0.000	0.087	0.962	0.964	0.977
Model 1	798.650	12	66.554	0.000	0.157	0.857	0.859	0.922
Model 2	231.853	12	19.321	0.000	0.083	0.959	0.961	0.975
Model 3	213.842	11	19.440	0.000	0.083	0.962	0.964	0.977
Model 4	806.800	13	62.062	0.000	0.151	0.856	0.858	0.920
Model 5	1457.708	14	104.122	0.000	0.197	0.739	0.741	0.834
United States								
Model	Δχ^2^	Δdf	χ^2^/df	*p*	RMSEA	NFI	CFI	GFI
Reference model	159.103	10	15.910	0.000	0.092	0.946	0.949	0.976
Model 1	425.722	12	35.477	0.000	0.140	0.855	0.858	0.937
Model 2	163.051	12	13.588	0.000	0.085	0.944	0.948	0.975
Model 3	158.211	11	14.474	0.000	0.088	0.946	0.949	-
Model 4	429.669	13	33.051	0.000	0.135	0.853	0.857	0.937
Model 5	502.138	14	35.867	0.000	0.141	0.829	0.832	0.920
England (United Kingdom)								
Model	Δχ^2^	Δdf	χ^2^/df	*p*	RMSEA	NFI	CFI	GFI
Reference model	215.012	10	21.501	0.000	0.112	0.934	0.936	0.964
Model 1	407.445	12	33.954	0.000	0.142	0.874	0.877	0.934
Model 2	225.503	12	18.792	0.000	0.104	0.930	0.934	0.963
Model 3	215.057	11	19.551	0.000	0.106	0.934	0.937	0.964
Model 4	407.446	13	31.342	0.000	0.136	0.874	0.878	0.934
Model 5	448.670	14	32.000	0.000	0.137	0.862	0.865	0.923
Australia								
Model	Δχ^2^	Δdf	χ^2^/df	*p*	RMSEA	NFI	CFI	GFI
Reference model	253.689	10	25.369	0.000	0.102	0.937	0.939	0.970
Model 1	534.465	12	44.539	0.000	0.137	0.868	0.870	0.938
Model 2	256.097	12	21.341	0.000	0.093	0.937	0.939	0.969
Model 3	253.786	11	23.071	0.000	0.097	0.937	0.940	0.970
Model 4	534.591	13	41.122	0.000	0.131	0.868	0.870	0.938
Model 5	673.026	14	48.073	0.000	0.142	0.833	0.836	0.917

Notes: Benchmark model: A four-factor model of distributed school leadership, teachers’ time spent participating in school leadership, teachers’ job stress, and teachers’ job satisfaction; Model 1: A three-factor model of distributed school leadership + teachers’ job stress, proportion of teachers’ time spent participating in school leadership, and teachers’ job satisfaction; Model 2: A three-factor model of teachers’ time spent participating in school leadership + teachers’ work stress, distributed school leadership, and teachers’ job satisfaction; Model 3: A three-factor model of distributed school leadership + teachers’ time spent participating in school leadership, teachers’ job stress, and teachers’ job satisfaction; Model 4: A two-factor model of distributed school leadership + proportion of teachers’ time spent participating in school leadership + teachers’ job stress and teachers’ job satisfaction; Model 5: A single-factor model, where all variables were combined into one factor.

**Table 5 behavsci-14-00279-t005:** Comparison of adjusted measurement models in four countries.

Shanghai (China)								
Model	Δχ^2^	Δdf	χ^2^/df	*p*	GFI	CFI	NFI	RMSEA
Reference model	202.555	7	28.936	0.000	0.974	0.965	0.964	0.102
Model 1	778.676	8	97.334	0.000	0.912	0.861	0.860	0.190
Model 2	1434.792	9	159.421	0.000	0.813	0.743	0.742	0.244
United States								
Model	Δχ^2^	Δdf	χ^2^/df	*p*	GFI	CFI	NFI	RMSEA
Reference model	154.439	7	22.063	0.000	0.972	0.949	0.947	0.109
Model 1	421.502	8	52.688	0.000	0.928	0.858	0.856	0.172
Model 2	494.439	9	54.938	0.000	0.910	0.833	0.831	0.175
England (United Kingdom)								
Model	Δχ^2^	Δdf	χ^2^/df	*p*	GFI	CFI	NFI	RMSEA
Reference model	168.452	7	24.065	0.000	0.968	0.949	0.947	0.118
Model 1	351.417	8	43.927	0.000	0.933	0.892	0.890	0.162
Model 2	392.984	9	43.665	0.000	0.920	0.879	0.877	0.161
Australia								
Model	Δχ^2^	Δdf	χ^2^/df	*p*	GFI	CFI	NFI	RMSEA
Reference model	215.953	7	30.850	0.000	0.970	0.948	0.946	0.113
Model 1	494.409	8	61.801	0.000	0.934	0.878	9.876	0.161
Model 2	632.834	9	70.315	0.000	0.909	0.843	0.842	0.172

Notes: Benchmark model: A three-factor model of distributed school leadership, teachers’ job stress, and teachers’ job satisfaction; Model 1: A three-factor model of distributed school leadership + teachers’ job stress and teachers’ job satisfaction; Model 2: A single-factor model, where all variables were combined into one factor.

**Table 6 behavsci-14-00279-t006:** Direct effect, indirect effect, and total effect of distributed school leadership on teachers’ job satisfaction.

Shanghai (China)					
	Effect size	SE	*p*	95%CI	
				Lower	Upper
Direct	0.385	0.020	0.000	0.347	0.426
Indirect	0.106	0.012	0.000	0.084	0.130
Total	0.492	0.022	0.000	0.450	0.536
United States					
	Effect size	SE	*p*	95%CI	
				Lower	Upper
Direct	0.938	0.062	0.000	0.818	1.061
Indirect	0.342	0.048	0.000	0.253	0.443
Total	1.280	0.069	0.000	1.149	1.421
England (United Kingdom)					
	Effect size	SE	*p*	95%CI	
				Lower	Upper
Direct	0.313	0.021	0.000	0.273	0.354
Indirect	0.214	0.021	0.000	0.176	0.256
Total	0.527	0.023	0.000	0.484	0.575
Australia					
	Effect size	SE	*p*	95%CI	
				Lower	Upper
Direct	0.278	0.016	0.000	0.247	0.310
Indirect	0.141	0.015	0.000	0.114	0.172
Total	0.420	0.017	0.000	0.386	0.455

## Data Availability

The data are available at https://www.oecd.org/education/talis/talis-2018-data.htm (accessed on 19 December 2023).

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
