# Peer review of "The Chain Mediating Role of Teachers’ Job Stress in the Influence of Distributed School Leadership on Job Satisfaction: Evidence from China, the United States, England, and Australia"

_behavsci, 2024, doi:10.3390/bs14040279_

Round 1

Reviewer 1 Report

Comments and Suggestions for Authors

This work is interesting but needs major revision.

While the abstract mentions a mediating effect of job stress, it does not present evidence of a "chain mediating" effect as the title implies.

In fact, the authors report only teachers' job stress as a mediator parameter; thus, the term "chain mediating effect" is inappropriate here, as there are no multiple mediators involved in a causal chain mediating effect on the dependent variable.

 The manuscript oversimplified the issue of distributed leadership styles. The authors could explain how a leader can exhibit characteristics of multiple leadership styles. A school principal, for example, could use distributed leadership by involving teachers in decision-making, while also demonstrating transformational leadership by motivating students and staff to excel academically. Different leadership styles can exhibit distributed leadership traits such as decision-making participation and collaboration with their Principal.

 In the introduction, the authors could include recent research on teacher stress and identify areas not thoroughly explored in your existing literature review, include studies on the impact of economic hardship, working conditions, age, and personality traits on teachers' stress levels.

 The authors should acknowledge that while effective leadership plays a crucial role in teacher job satisfaction, other external factors also significantly impact job satisfaction and occupational stress.  These can include economic conditions, workload, age, experience and personality traits. Therefore, a leader's influence on teacher well-being has limitations, as evidenced by research highlighting the independent contributions of factors like economic hardship and age to teacher stress and satisfaction.

Another issue raised in this manuscript is how much power leaders will have in a highly centralized educational system, which is characterized by a centralized top-down decision and communication system; the authors should clarify whether Chinese schools differ in this regard from the schools in the other countries included in the present work.

 The authors could also clarify that job stress is influenced by various aspects beyond leadership style and highlight this in the limitations section and suggest that further research to explore the interplay between leadership and other factors like economic conditions or personality traits on teacher stress.

Author Response

Thank you so much for your professional and insightful comments and we really appreciate your comprehensive and professional comments for us to promote the quality of this manuscript. During the revising process, we have learned a lot from your persistence, patience, and profession. We really appreciate your valuable guidance and intelligent ideas! Based on your comments and suggestion, the revision has been completed as shown as below:

Reviewer 1

Q1: This work is interesting but needs major revision.
While the abstract mentions a mediating effect of job stress, it does not present evidence of a "chain mediating" effect as the title implies. In fact, the authors report only teachers' job stress as a mediator parameter; thus, the term "chain mediating effect" is inappropriate here, as there are no multiple mediators involved in a causal chain mediating effect on the dependent variable.

Response: Thank you so much for your comments and following your comments, we have revised, in the abstract, the expression of "chain mediation" of two variables is modified and the revision is provided as follows: “Distributed leadership has been shown to improve teacher job satisfaction and reduce teacher job stress. However, few studies have thoroughly considered the indirect effects of distributed leadership on increasing teachers' time in school administration and management, thereby increasing work stress and thereby decreasing job satisfaction.” (p-1).

Q2: The manuscript oversimplified the issue of distributed leadership styles. The authors could explain how a leader can exhibit characteristics of multiple leadership styles. A school principal, for example, could use distributed leadership by involving teachers in decision-making, while also demonstrating transformational leadership by motivating students and staff to excel academically. Different leadership styles can exhibit distributed leadership traits such as decision-making participation and collaboration with their Principal.

Response: Thank you so much for your professional suggestion and actually the leadership style of principals was considered and considering that it only focused on the level of teachers and did not focus on the level of principals (i.e. leadership style) or school (i.e. leadership atmosphere), it did not discuss the above content too much, nor did it discuss the influence of different leadership styles on teachers' behavior. Leadership style and leadership climate are indeed important things to pay attention to, but they are not emphasized in this study, so this study will explain them. In addition, following your suggestions, we have discussed it in the revision as follows: “TALIS survey defines the actual performance of distributed leadership as the opportunity to participate in school decision-making, the need to take responsibility for school affairs, and the ability to feel the cooperative cultural atmosphere. This is the degree of distributed leadership in teachers' subjective perception. It is not an objective principal leadership style or distributed leadership atmosphere on campus.” (p.2).

Q3: In the introduction, the authors could include recent research on teacher stress and identify areas not thoroughly explored in your existing literature review, include studies on the impact of economic hardship, working conditions, age, and personality traits on teachers' stress levels.

Response: Sure. We totally agree with your insightful ideas to add more content to explore recent research on teacher stress and identify areas and the revision is provided as follows: “Teachers’ job stress derives from various factors, including economic hardship, working conditions, age, and personality traits on teachers' stress levels (Anh-Duc, 2023; Aryawan et al., 2024; Boyaci et al., 2018; Darlene, 2018). For example, according to the RAND Corporation's "Teacher Happiness and Resignation Intention", although the level of job stress of US teachers has recovered somewhat after the COVID-19 pandemic, the hap-piness of teachers is still not high compared with other workers. The main sources of job stress for US teachers are managing student behavior (46%), supporting students' aca-demic learning (34%) and administrative work (29%). About a quarter of teachers report that administrative work, low pay, and long hours are the biggest job-related stressors (Doan et al., 2023). In November of the same year, Education Support, a British Teacher welfare charity, released the Teacher Wellbeing Index 2023, which pointed out that the happiness of British teachers was at its lowest level since 2019, surveying more than 3,004 educators. 78% reported burnout, up 3 percentage points from 2022, and 36% reported burnout, up 9 percentage points from 2022, with loneliness and social isolation as a source of stress (Education Support, 2023). According to the Mental Health Foundation of Aus-tralia, more than 50 per cent of Australian teachers suffer from anxiety and almost one in five suffer from depression (Geale, 2022). It can be said that the increasing work pressure of teachers has become a reality in different countries (Salazar, 2023). The high work pressure caused by low income and heavy workload reduces the happiness of teachers and increases the turnover rate of teachers. Delegating power to teachers, adjusting workload freely and relieving work pressure are the fundamental ways to increase teachers' job satisfaction (Hwang et al., 2022).” (p.1-2).

 Q4: The authors should acknowledge that while effective leadership plays a crucial role in teacher job satisfaction, other external factors also significantly impact job satisfaction and occupational stress.  These can include economic conditions, workload, age, experience and personality traits. Therefore, a leader's influence on teacher well-being has limitations, as evidenced by research highlighting the independent contributions of factors like economic hardship and age to teacher stress and satisfaction.

Response: We totally agree with your insight comments and many thanks for your professional comments and suggestions. Following your suggestions, we have added the content regarding to examining the teacher job satisfaction, other external factors also significantly impact job satisfaction and occupational stress in the introduction section and also discussed it in the limitation section and the revision is provided as follows: “It relates to the sense of accomplishment and achievement generated by everyday ac-tivities of the profession (Collie et al., 2012; Klassen&Chiu, 2010). There are many kinds of external factors that also significantly impact job satisfaction and occupational stress. For example, Din-ham and Scott (1998) classify the sources of teacher job satisfaction into "core business of teaching" factors (student achievement, professional self-growth); School level factors (school leadership, climate, decision making, school infrastructure, school reputation); And system-level/social factors (workload and impact of change; The status and image of teachers; Promotion results). The principal's behavior creates a different working environment in the school, which in turn affects the teacher's satisfaction.” (p.3).

Q5:  Another issue raised in this manuscript is how much power leaders will have in a highly centralized educational system, which is characterized by a centralized top-down decision and communication system; the authors should clarify whether Chinese schools differ in this regard from the schools in the other countries included in the present work.

Response: We really appreciate your valuable and enlightened suggestions and totally agree with your insight comments and many thanks for your professional comments and suggestions. Following your suggestions, we have added more content regarding to clarify whether Chinese schools differ in this regard from the schools in the other countries in the discussion section and the revision is provided as follows: “Although active participation in school leadership increases the proportion of non-teaching time, the impact on work stress is not significant, which can be explained by teachers' work initiative and enthusiasm, it can also be explained by the low proportion of school leadership time in the four countries (the average value is around 10%). Ordinary teachers do not really participate in school leaders at ordinary times, and their energy consumption is insignificant compared with lesson preparation, teaching and research, homework correction and communication with parents, which is not enough to make ordinary teachers really feel pressure, or even have a direct impact on job satisfaction. Therefore, in the model, there is no intermediary role (Li et al. (2021), Torres (2019), Hall (2013), and Christine (2019).

It is found that the selected four countries have in common, but the model ultimately reveals differences among the four countries. For example, the results of the study show that distributed leadership in Shanghai and England has a significant positive impact on the amount of time teachers spend in school leadership, while the United States and Australia do not. The education systems in England and China are more centralized than those in the other two countries, with a tradition of top-down decision-making and communication in education management. Distributed leadership is indeed an obvious and mandatory requirement for teachers to become school leaders, and teachers' identity increases, their responsibilities increase, and their working hours increase significantly. In Shanghai and the United States, the proportion of teachers' time spent in school leadership has a positive impact on teacher job satisfaction, while in England and Australia it has a negative impact on teacher job satisfaction. This reflects the two different teacher work cultures, which believe that "more effort will have more return". In Shanghai and Australia, the proportion of teachers' time spent in school leadership has a positive impact on teachers' stress, while in Britain and the United States, it has a negative impact on teachers' stress, which reflects that teachers in Britain and the United States may have a higher ability to withstand pressure or have more love for the teaching profession.” (p.14).

Q6: The authors could also clarify that job stress is influenced by various aspects beyond leadership style and highlight this in the limitations section and suggest that further research to explore the interplay between leadership and other factors like economic conditions or personality traits on teacher stress.

Response: Sure. many thanks for your professional comments and suggestions. Following your suggestions, we have added the content in regard to exploring various aspects beyond leadership style and highlight this in the limitations section and the revision is provided as follows: “Fourthly, in addition to teacher leadership behavior, other external factors, including exogenous factors such as economy, education policy and workload, as well as endoge-nous factors such as gender, teaching experience and organizational commitment, can also significantly affect teachers' job satisfaction and work stress through literature review. These factors are not considered in the summary of this study. The mechanism of these factors can be further discussed in the future. Fifth, in terms of analysis methods, this study mainly adopts AMOS path analysis, and does not deeply explore the mediating role of variables (and possible moderating role).” (p5)。

Reviewer 2 Report

Comments and Suggestions for Authors

I read the article with great interest. The authors show deep knowledge of the relevant literature and have proposed a solid approach to verify the assumptions. Yet there are some things in the article that could be improved, as they would help the article reach a broader audience.

1. The description of the theoretical path models. The authors propose a theoretical model which is based on literature. Yet the definition of the relation between the variables is not quite clear (line 148). I would suggest adding symbols how for example disturbed leadership impacts job satisfaction (positively -> marked with a plus over the arrow), and how it impacts for example stress (also a positive relation, thus also a plus). But then the relation between stress and job satisfaction is negative, so a minus over that arrow.

2. The description of the analysed path models - I had trouble locating the description of the way in which the authors constructed the models 1-5 for the path analysis (Table 3). If there is one theoretical model proposed, so should one model be analysed. If authors see different alternative theoretical models that could relate the constructs, they should propose them before commencing to analyses. Now it looks like the authors failed to confirm the initial model, so they started to search for alternative ones.

3. Cross-cultural invariance analysis is missing. As authors use the same scales applied in different countries, they should check if the meaning of the answered questions is the same in all countries. Only when a certain level of invariance (scalar) is reached, the countries can be meaningfully compared). See for example Miller and Sheu, 2008 for reference. 

4. I believe on type of analysis should be sufficient - in the analysed analysis it should be either path analysis with AMOS (where authors are close to showing nice, multicultural examples) OR the subsequent med-mod analysis (where the authors show less doubt about the results, yet fail to unify the countries.

Addressing the abovementioned issues can help to transfer this good article into a great article/

Author Response

Reviewer 2

I read the article with great interest. The authors show deep knowledge of the relevant literature and have proposed a solid approach to verify the assumptions. Yet there are some things in the article that could be improved, as they would help the article reach a broader audience.

  1. The description of the theoretical path models. The authors propose a theoretical model which is based on literature. Yet the definition of the relation between the variables is not quite clear (line 148). I would suggest adding symbols how for example disturbed leadership impacts job satisfaction (positively -> marked with a plus over the arrow), and how it impacts for example stress (also a positive relation, thus also a plus). But then the relation between stress and job satisfaction is negative, so a minus over that arrow.

Response: We really appreciate your valuable and enlightened suggestions and totally agree with your insight comments and many thanks for your professional comments and suggestions. Following your suggestions, the description of the theoretical path models has been revised and the revision is offered as follows:

 Figure 1. Hypothesized relationship path diagram. Note: + indicates a positive impact, - indicates a negative impact, and +/- indicates an uncertain impact direction

2.The description of the analysed path models - I had trouble locating the description of the way in which the authors constructed the models 1-5 for the path analysis (Table 3). If there is one theoretical model proposed, so should one model be analysed. If authors see different alternative theoretical models that could relate the constructs, they should propose them before commencing to analyses. Now it looks like the authors failed to confirm the initial model, so they started to search for alternative ones.

Response: Thank you so much for your professional suggestion and Models 1-5 in Table 3 are the discriminative validity of the analysis variables, rather than the final theoretical model, in order to show that the four variables belong to different factors. However, for TALIS data, "duration of teachers' participation in school management and administrative affairs" is an explicit variable, and only one blank filling question seriously affects the results of the discrimination validity analysis. At the same time, it is uncertain whether this explicit variable plays a mediating role between distributed leadership and teachers' job satisfaction, so a model without this explicit variable is also implied in the theoretical model (after clear processing). Therefore, this study re-analyzes the discriminative validity of the three variables after the exclusion of this explicit variable, and then proves that the other three latent variables except this explicit variable have good discriminative validity. Later, in the process of hypothesis testing, this study still adopted the most basic model of four variables, built model a-c, and verified the hypothesis. This recommendation indicates that the article needs more explanation to clear the reader's confusion about the model changes when testing the discriminative validity, so more explanation is added in the study.

  1. Cross-cultural invariance analysis is missing. As authors use the same scales applied in different countries, they should check if the meaning of the answered questions is the same in all countries. Only when a certain level of invariance (scalar) is reached, the countries can be meaningfully compared). See for example Miller and Sheu, 2008 for reference.

Response: We really appreciate insightful ideas and thank you so much for your professional suggestion and The TALIS 2018 survey conducted a measurement invariance analysis for different countries/economies, and the results were published in the TALIS 2018 Technical Report. This study supplemented the description of invariance results in the second part to make the variable description more complete. At the same time, based on the cleaned data, invariance analysis was conducted for two variables/dimensions not covered in the technical report, and the result showed that several variables could be used for correlation and linear regression-based analysis. And the revision is provided as follows:

2.3Cross-cultural invariance

In cross-cultural research, scales of different variables are translated into many versions to test subjects from different countries, so it is necessary to determine whether the scales of variables are consistent in different measurements through measurement invariance analysis (Miller et al. 2008). The data source for this study is a survey conducted by TALIS in 2018. The results of the cross-country measurement invariance of the questionnaire are published in TALIS2018 Technical Report of OECD, and the invariance hierarchy is divided into three levels (Chen, 2007) : configural level (the factor loadings and intercepts are allowed to vary across participating countries/economies, Further analysis of cross-country comparisons can only be carried out at the conceptual level), metric level (the score of the scale is created respectively with equal factor loadings but with intercepts allowed to vary across participating coun-tries/economies, which can be used for correlation and linear regression based analysis) and scalar level (which can compare the average size of participating countries /economies). Two dimensions of job stress and two dimensions of job satisfaction were included. At the same time, Mplus software was used to analyze the measurement invariance of distributed leadership variables and stakeholder dimensions of job stress, and the same rating method was used in the technical report. The Invariance level analysis results of each dimension of the final three variables are shown in the table below. Although each variable does not reach the scalar level, correlation and linear regression analysis can be carried out (Table 2).

Table 2. Invariance test results for each scale

Scale label

Invariance level

RMSEA

CFI

TLI

SRMR

ΔCFI

ΔTLI

Job satisfaction with work environment, teacher

Metric

0.056

0.969

0.962

0.092

0.017

-0.005

Job satisfaction with profession, teacher

Metric

0.071

0.961

0.94

0.088

0.037

0.049

Workload stress

Metric

0.068

0.949

0.936

0.069

0.027

0.003

Workplace stress

Metric

0.088

0.948

0.936

0.09

0.035

0.012

Stakeholder behavioural stress

Metric

0.079

0.958

0.94

0.089

0.035

0.015

Distributed leadership

Metric

0.093

0.995

0.998

0.044

0.025

0.023

  1. I believe on type of analysis should be sufficient - in the analysed analysis it should be either path analysis with AMOS (where authors are close to showing nice, multicultural examples) OR the subsequent med-mod analysis (where the authors show less doubt about the results yet fail to unify the countries.

Addressing the abovementioned issues can help to transfer this good article into a great article.

Response: Many thanks for your professional comments and suggestions. Following your suggestions, we have added the content the concerns and discussion in the limitation section and thank you so much for your carefully and comprehensive comments for us to promote the quality of this study and completely appreciate it.

Round 2

Reviewer 1 Report

Comments and Suggestions for Authors

Thank you for your effort to revise your manuscript according to my comments and suggestions.  I only have one issue, it is about the text in line 488.  I can see what you try to say , but in terms of leadership schools in Englad have more autonomy.  England's education system leans more towards decentralization in terms of school management and pedagogy.  This makes your arguments not valid.  Please revise this sentence.

Author Response

Q1: Thank you for your effort to revise your manuscript according to my comments and suggestions.  I only have one issue, it is about the text in line 488.  I can see what you try to say , but in terms of leadership schools in Englad have more autonomy.  England's education system leans more towards decentralization in terms of school management and pedagogy.  This makes your arguments not valid.  Please revise this sentence.

Response: Thank you so much for your carefully review and we really appreciate your comments and following your suggestions, we have revised and deleted the sentence regarding to the leadership schools in England and the revision is provided as follows:

“The education systems in England and China are more centralized than those in the other two countries, with a tradition of top-down decision-making and communication in education management.”